# REFACTOR GNNs: Revisiting Factorisation-based Models from a Message-Passing Perspective

**Yihong Chen**[ϒδ]    **Pushkar Mishra**[δ]    **Luca Franceschi**[Π]    **Pasquale Minervini**[ϒΩ]
**Pontus Stenetorp**[ϒ]    **Sebastian Riedel**[ϒδ]

[ϒ]UCL Centre for Artificial Intelligence, London, United Kingdom
[δ]Meta AI, London, United Kingdom
[Π]Amazon Web Services, Berlin, Germany
[Ω]School of Informatics, University of Edinburgh, Edinburgh, United Kingdom
{yihong.chen, p.stenetorp, s.riedel}@cs.ucl.ac.uk
pushkarmishra@meta.com   franuluc@amazon.de   p.minervini@ed.ac.uk

## Abstract

Factorisation-based Models (FMs), such as DistMult, have enjoyed enduring success for Knowledge Graph Completion (KGC) tasks, often outperforming Graph Neural Networks (GNNs). However, unlike GNNs, FMs struggle to incorporate node features and generalise to unseen nodes in inductive settings. Our work bridges the gap between FMs and GNNs by proposing REFACTOR GNNs. This new architecture draws upon *both* modelling paradigms, which previously were largely thought of as disjoint. Concretely, using a message-passing formalism, we show how FMs can be cast as GNNs by reformulating the gradient descent procedure as message-passing operations, which forms the basis of our REFACTOR GNNs. Across a multitude of well-established KGC benchmarks, our REFACTOR GNNs achieve comparable transductive performance to FMs, and state-of-the-art inductive performance while using an order of magnitude fewer parameters.

## 1 Introduction

In recent years, machine learning on graphs has attracted significant attention due to the abundance of graph-structured data and developments in graph learning algorithms. Graph Neural Networks (GNNs) have shown state-of-the-art performance for many graph-related problems, such as node classification [1] and graph classification [2]. Their main advantage is that they can easily be applied in an inductive setting: generalising to new nodes and graphs without re-training. However, despite many attempts at applying GNNs for multi-relational link prediction tasks such as Knowledge Graph Completion [3], there are still few positive results compared to factorisation-based models (FMs) [4, 5]. As it stands, GNNs either – after resolving reproducibility concerns – deliver significantly lower performance [6, 7] or yield negligible performance gains at the cost of highly sophisticated architecture designs [8]. A notable exception is NBFNet [9], but even here improvements come at the price of high computational inference costs compared to FMs. Furthermore, it is unclear how NBFNet could incorporate node features, which – as we will see in this work – leads to remarkably lower performance in an inductive setting. On the other hand FMs, despite being a simpler architecture, have been found to be very accurate for knowledge graph completion when coupled with appropriate training strategies [10] and training objectives [11, 12]. However, they also come with shortcomings in that they, unlike GNNs, can not be applied in an inductive setting.

Given the respective strengths and weaknesses of FMs and GNNs, *can we bridge these two seemingly different model categories?*. In this work, we first show that we can build the link between the two by identifying the implicit message-passing within FMs. Later in our appendix, we showcase a direct application of this link - inductivising FMs by proposing REFACTOR GNNs.

Y. Chen et al., REFACTOR GNNs: Revisiting Factorisation-based Models from a Message-Passing Perspective (Extended Abstract). Presented at the First Learning on Graphs Conference (LoG 2022), Virtual Event, December 9–12, 2022.

## 2 Background

Knowledge Graph Completion (KGC) [13] is a canonical task of multi-relational link prediction. The goal is to predict missing edges given the existing edges in the knowledge graph. Formally, a knowledge graph contains a set of entities (nodes) $\mathcal{E} = \{1, \ldots, |\mathcal{E}|\}$, a set of relations (edge types) $\mathcal{R} = \{1, \ldots, |\mathcal{R}|\}$, and a set of typed edges between the entities $\mathcal{T} = \{(v_i, r_i, w_i)\}_{i=1}^{|\mathcal{T}|}$, where each triplet $(v_i, r_i, w_i)$ indicates a relationship of type $r_i \in \mathcal{R}$ between the *subject* $v_i \in \mathcal{E}$ and the *object* $w_i \in \mathcal{E}$. Given a node $v$, we denote its *outgoing* 1-hop neighbourhood as the set of relation-object pairs $\mathcal{N}_+^1[\mathrm{v}] = \{(r, o) \mid (\mathrm{v}, r, o) \in \mathcal{T}\}$, its *incoming* 1-hop neighbourhood as the set of subject-relation pairs $\mathcal{N}_-^1[\mathrm{v}] = \{(r, s) \mid (s, r, \mathrm{v}) \in \mathcal{T}\}$, and $\mathcal{N}^1[v] = \mathcal{N}_+^1[v] \cup \mathcal{N}_-^1[v]$ the union of the two. We denote the neighbourhood of $v$ under a specific relation $r$ as $\mathcal{N}_\pm^1[r, v]$. Entities may come with features $X \in \mathbb{R}^{|\mathcal{E}| \times K}$ for describing them, such as textual encodings of their names and/or descriptions. Multi-relational link prediction models can be trained via maximum likelihood, by fitting a parameterized conditional categorical distribution $P_\theta(w \mid v, r)$ over the candidate objects of a relation, given the subject $v$ and the relation type $r$:

$$P_\theta(w| \mathrm{v}, \mathrm{r}) = \frac{\exp \Gamma_\theta(\mathrm{v}, \mathrm{r}, w)}{\sum_{u \in \mathcal{E}} \exp \Gamma_\theta(\mathrm{v}, \mathrm{r}, u)} = \mathrm{Softmax}(\Gamma_\theta(\mathrm{v}, \mathrm{r}, \cdot))[w]. \tag{1}$$

Here $\Gamma_\theta : \mathcal{E} \times \mathcal{R} \times \mathcal{E} \to \mathbb{R}$ is a *scoring function*, which, given a triplet $(v, r, w)$, returns the likelihood that the corresponding edge appears in the knowledge graph. We illustrate our derivations using DistMult [4] as the score function $\Gamma$ and defer extensions to general score functions, e.g. ComplEx [5], to the appendix. For DistMult, the score function $\Gamma_\theta$ is defined as the tri-linear dot product of the vector representations corresponding to the subject, relation, and object of the triplet:

$$\Gamma_\theta(v, r, w) = \langle f_\phi(v), f_\phi(w), g_\psi(r) \rangle = \sum_{i=1}^{K} f_\phi(v)_i f_\phi(w)_i g_\psi(r)_i, \tag{2}$$

where $f_\phi : \mathcal{E} \to \mathbb{R}^K$ and $g_\psi : \mathcal{R} \to \mathbb{R}^K$ are learnable maps parameterised by $\phi$ and $\psi$ that encode entities and relation types into $K$-dimensional vector representations, and $\theta = (\phi, \psi)$. We will refer to $f$ and $g$ as the entity and relation *encoders*, respectively. If we define the data distribution as $P_D(x) = \frac{1}{|\mathcal{T}|} \sum_{(v,r,w) \in \mathcal{T}} \delta_{(v,r,w)}(x)$, where $\delta_{(v,r,w)}(x)$ is a Dirac delta function at $(v, r, w)$, then the objective is to learn the model parameters $\theta$ by minimising the expected negative log-likelihood $\mathcal{L}(\theta)$ of the ground-truth entities for the queries $(v, r, ?)$ obtained from $\mathcal{T}$:

$$\arg \min_\theta \mathcal{L}(\theta) \quad \text{where} \quad \mathcal{L}(\theta) = -\mathbb{E}_{x \sim P_D}[\log(P_\theta(w|v, r)] = -\frac{1}{|\mathcal{T}|} \sum_{(v,r,w) \in \mathcal{T}} \log P_\theta(w|v, r). \tag{3}$$

During inference, we use $P_\theta$ for determining the plausibility of links not present in the training graph.

### 2.1 Factorisation-based Models for KGC

In factorisation-based models, which we assume to be DistMult, $f_\phi$ and $g_\psi$ are simply parameterised as look-up tables, associating each entity and relation with a continuous distributed representation:

$$f_\phi(v) = \phi[v], \ \phi \in \mathbb{R}^{|\mathcal{E}| \times K} \quad \text{and} \quad g_\psi(r) = \psi[r], \ \psi \in \mathbb{R}^{|\mathcal{R}| \times K}. \tag{4}$$

### 2.2 GNN-based Models for KGC

GNNs were originally proposed for node or graph classification tasks [14, 15]. To adapt them to KGC, previous work has explored two different paradigms: *node-wise entity representations* [16] and *pair-wise entity representations* [9, 17]. Though the latter paradigm has shown promising results, it requires computing representations for all pairs of nodes, which can be computationally expensive for large-scale graphs with millions of entities. Additionally, node-wise representations allow for using a single evaluation of $f_\phi(v)$ for multiple queries involving $v$.

Models based on the first paradigm differ from pure FMs only in the entity encoder and lend themselves well for a fairer comparison with pure FMs. We will therefore focus on this class and leave the investigation of pair-wise representations to future work. Let $q_\phi : \mathcal{G} \times \mathcal{X} \to \bigcup_{S \in \mathbb{N}^+} \mathbb{R}^{S \times K}$ be a GNN encoder, where $\mathcal{G} = \{G \mid G \subseteq \mathcal{E} \times \mathcal{R} \times \mathcal{E}\}$ is the set of all possible multi-relational

graphs defined over $\mathcal{E}$ and $\mathcal{R}$, and $\mathcal{X}$ is the input feature space, respectively. Then we can set $f_\phi(v) = q_\phi(\mathcal{T}, X)[v]$. Following the standard message-passing framework [2, 18] used by the GNNs, we view $q_\phi = q^L \circ ... \circ q^1$ as the recursive composition of $L \in \mathbb{N}^+$ layers that compute intermediate representations $h^l$ for $l \in \{1, \ldots, L\}$ (and $h^0 = X$) for all entities in the KG. Each layer is made up of the following three functions:

1. A *message function* $q_{\mathrm{M}}^l : \mathbb{R}^K \times \mathcal{R} \times \mathbb{R}^K \to \mathbb{R}^K$ that computes the message along each edge. Given an edge $(v, r, w) \in \mathcal{T}$, $q_{\mathrm{M}}^l$ not only makes use of the node states $h^{l-1}[v]$ and $h^{l-1}[w]$ (as in standard GNNs) but also uses the relation $r$; denote the message as
$$m^l[v, r, w] = q_{\mathrm{M}}^l\left(h^{l-1}[v], r, h^{l-1}[w]\right);$$

2. An *aggregation function* $q_{\mathrm{A}}^l : \bigcup_{S \in \mathbb{N}} \mathbb{R}^{S \times K} \to \mathbb{R}^K$ that aggregates all messages from the 1-hop neighbourhood of a node; denote the aggregated message as
$$z^l[v] = q_{\mathrm{A}}^l\left(\{m^l[v, r, w] \mid (r, w) \in \mathcal{N}^1[v]\}\right);$$

3. An *update function* $q_{\mathrm{U}}^l : \mathbb{R}^K \times \mathbb{R}^K \to \mathbb{R}^K$ that produces the new node states $h^l$ by combining previous node states $h^{l-1}$ and the aggregated messages $z^l$:
$$h^l[v] = q_{\mathrm{U}}^l(h^{l-1}[v], z^l[v]).$$

Different parameterisations of $q_{\mathrm{M}}^l$, $q_{\mathrm{A}}^l$, and $q_{\mathrm{U}}^l$ lead to different GNNs. For example, R-GCNs [16] define the $q_{\mathrm{M}}^l$ function using per-relation the linear transformations $m^l[v, r, w] = \frac{1}{\mathcal{N}^1[r,v]} W_r^l h^{l-1}[w]$; $q_{\mathrm{A}}^l$ is implemented by a summation and $q_{\mathrm{U}}^l$ is a non-linear transformation $h^l[v] = \sigma(z^l[v] + W_0^l h^{l-1}[v])$, where $\sigma$ is the sigmoid function. For each layer, the learnable parameters are $\{W_r^l\}_{r \in \mathcal{R}}$ and $W_0^l$, all of which are matrices in $\mathbb{R}^{K \times K}$.

## 3 Implicit Message-Passing in FMs

The sharp difference in analytical forms might give rise to the misconception that GNNs incorporate message-passing over the neighbourhood of each node (up to $L$-hops), while FMs do not. In this work, we show that by explicitly considering the training dynamics of FMs, we can uncover and analyse the hidden message-passing mechanism within FMs. In turn, this will lead us to the formulation of a novel class of GNNs well suited for multi-relational link prediction tasks (Appendix A). Specifically, we propose to interpret the FMs' optimisation process of their objective (3) as the entity encoder. If we consider, for simplicity, a gradient descent training dynamic, then
$$f_{\phi^t}(v) = \phi^t[v] = \mathrm{GD}^t(\phi^{t-1}, \mathcal{T})[v] = \underbrace{\mathrm{GD}^t \circ ... \mathrm{GD}^1}_{t}(\phi^0, \mathcal{T})[v], \tag{5}$$

where $\phi^t$ is the embedding vector at the $t$-th step, $t \in \mathbb{N}^+$ is the total number of iterations and $\phi^0$ is a random initialisation. GD is the gradient descent operator:
$$\mathrm{GD}(\phi, \mathcal{T}) = \phi - \alpha \nabla_\phi \mathcal{L} = \phi + \alpha \sum_{(v,r,w) \in \mathcal{T}} \frac{\partial \log P(w|v, r)}{\partial \phi}, \tag{6}$$

where $\alpha = \beta |\mathcal{T}|^{-1}$, with a $\eta > 0$ learning rate. We now dissect Equation (6) in two different (but equivalent) ways. In the first, which we dub the *edge view*, we separately consider each addend of the gradient $\nabla_\phi \mathcal{L}$. In the second, we aggregate the contributions from all the triplets to the update of a particular node. With this latter decomposition, which we call the *node view*, we can explicate the message-passing mechanism at the core of the FMs. While the edge view suits a vectorised implementation better, the node view further exposes the information flow among nodes, allowing us to draw an analogy to message-passing GNNs.

### 3.1 The Edge View

Each addend of Equation (6) corresponds to a single edge $(v, r, w) \in \mathcal{T}$ and contributes to the update of the representation of all nodes. The update on the representation of the subject $\phi[v]$ is:

$$\mathrm{GD}(\phi, \{(v, r, w)\})[v] = \phi[v] + \alpha \left( \underbrace{g(r) \odot \phi[w]}_{w \to v} - \underbrace{\sum_{u \in \mathcal{E}} P_\theta(u|v, r) g(r) \odot \phi[u]}_{u \to v} \right).$$

The $w \to v$ term indicates information flow from $w$ (a neighbour of $v$) to $v$, increasing the score of the gold triplet $(v, r, w)$. The $u \to v$ term indicates information flow from global nodes, decreasing the scores of triplets $(v, r, ?)$ with $v$ as the subject and $r$ as the predicate. Similarly, for the object $w$,

$$\text{GD}(\phi, \{(v, r, w)\})[w] = \phi[w] + \alpha \underbrace{(1 - P_\theta(w|v, r))\, g(r) \odot \phi[v]}_{v \to w},$$

where, again, the $v \to w$ term indicates information flow from the neighbouring node $v$. Finally, for the nodes other than $v$ and $w$, we have

$$\text{GD}(\phi, \{(v, r, w)\})[u] = \phi[u] + \alpha \left( \underbrace{-P_\theta(u|v, r)\phi[v] \odot g(r)}_{v \to u} \right).$$

## 3.2   The Node View

To fully uncover the message-passing mechanism of FMs, we now focus on the gradient descent operation over a single node $v \in \mathcal{E}$, referred to as the *central node* in the GNN literature. Recalling Equation (6), we have:

$$\text{GD}(\phi, \mathcal{T})[v] = \phi[v] + \alpha \sum_{(v, r, w) \in \mathcal{T}} \frac{\partial \log P(\bar{\text{w}} \,|\, \bar{\text{v}}, \bar{\text{r}})}{\partial \phi[v]}, \tag{7}$$

which aggregates the information stemming from the updates presented in the edge view. The next theorem describes how this total information flow to a particular node can be recast as an instance of message passing (cf. Section 2.2). We defer the proof to the appendix.

**Theorem 3.1** (Message passing in FMs). *The gradient descent operator* $\text{GD}$ *(Equation (7)) on the node embeddings of a DistMult model (Equation (4)) with the maximum likelihood objective in Equation (3) and a multi-relational graph* $\mathcal{T}$ *defined over entities* $\mathcal{E}$ *induces a message-passing operator whose composing functions are:*

$$q_\text{M}(\phi[v], r, \phi[w]) = \begin{cases} \phi[w] \odot g(r) & \text{if } (r, w) \in \mathcal{N}_+^1[v], \\ (1 - P_\theta(v|w, r))\phi[w] \odot g(r) & \text{if } (r, w) \in \mathcal{N}_-^1[v]; \end{cases} \tag{8}$$

$$q_\text{A}(\{m[v, r, w] \,:\, (r, w) \in \mathcal{N}^1[v]\}) = \sum_{(r, w) \in \mathcal{N}^1[v]} m[v, r, w]; \tag{9}$$

$$q_\text{U}(\phi[v], z[v]) = \phi[v] + \alpha z[v] - \beta n[v], \tag{10}$$

*where, defining the sets of triplets* $\mathcal{T}^{-v} = \{(s, r, o) \in \mathcal{T} \,:\, s \neq v \wedge o \neq v\}$,

$$n[v] = \frac{|\mathcal{N}_+^1[v]|}{|\mathcal{T}|} \mathbb{E}_{P_{\mathcal{N}_+^1[v]}} \mathbb{E}_{u \sim P_\theta(\cdot|v, r)} g(r) \odot \phi[u] + \frac{|\mathcal{T}^{-v}|}{|\mathcal{T}|} \mathbb{E}_{P_{\mathcal{T}^{-v}}} P_\theta(v|s, r) g(r) \odot \phi[s], \tag{11}$$

*where* $P_{\mathcal{N}_+^1[v]}$ *and* $P_{\mathcal{T}^{-v}}$ *are the empirical probability distributions associated to the respective sets.*

What emerges from the equations is that each GD step contains an explicit information flow from the neighbourhood of each node, which is then aggregated with a simple summation. Through this direct information path, $t$ GD steps cover the $t$-hop neighbourhood of $v$. As $t$ goes towards infinity – or in practice – as training converges, FMs capture the global graph structure. The update function (10) somewhat deviates from classic message passing frameworks as $n[v]$ of Equation (11) involves global information. However, we note that we can interpret this mechanism under the framework of augmented message passing [19] and, in particular, as an instance of *graph rewiring*.

Based on Theorem 3.1 and Equation (5), we can now view $\phi$ as the transient node states $h$ (cf. Section 2.2) and GD on node embeddings as a message-passing layer. This dualism sits at the core of our proposed REFACTOR GNN model, which we will leave to the appendix due to limited space. Intuitively, REFACTOR GNNS inductivise FMs via simply truncating the infinite message-passing into a finite one. Across a multitude of well-established KGC benchmarks, REFACTOR GNNS achieve comparable transductive performance to FMs, and state-of-the-art inductive performance while using an order of magnitude fewer parameters.

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

## A REFACTOR GNNs

FMs are trained by minimising the objective (3), initialising both sets of parameters ($\phi$ and $\psi$) and performing GD until approximate convergence (or until early stopping terminates the training). The implications are twofold: $i$) the initial value of the entity lookup table $\phi$ does not play any major role in the final model after convergence, and $ii$) if we introduce a new set of entities, the conventional wisdom is to retrain[1] the model on the expanded knowledge graph. This is computationally rather expensive compared to the "inductive" models that require no additional training and can leverage node features like entity descriptions. However, as we have just seen in Theorem 3.1, the training procedure of FMs may be naturally recast as a message-passing operation, which suggests that it is possible to use FMs for inductive learning tasks. In fact, we envision that there is an entire novel spectrum of model architectures interpolating between pure FMs and (various instantiations of) GNNs. Here we propose one simple implementation of such an architecture which we dub REFACTOR GNNs. Figure 1 gives an overview of REFACTOR GNNs.

**The REFACTOR Layer.** A REFACTOR GNN contains $L$ REFACTOR layers, that we derive from Theorem 3.1. Aligning with the notations in Section 2.2, given a KG $\mathcal{T}$ and entity representations $h^{l-1} \in \mathbb{R}^{|\mathcal{E}| \times K}$, the REFACTOR layer computes the representation of a node $v$ as follows:

$$h^l[v] = q^l(\mathcal{T}, h^{l-1})[v] = h^{l-1}[v] - \beta n^l[v] + \alpha \sum_{(r,w) \in \mathcal{N}^1[v]} q^l_M(h^{l-1}[v], r, h^{l-1}[w]), \qquad (12)$$

where the terms $n^l$ and $q^l_M$ are derived from Equation (11) and Equation (8), respectively. We note that REFACTOR GNNs treat incoming and outgoing neighbourhoods differently instead of treating them equally as in the R-GCN, the first GNN on multi-relational graphs [16].

Equation (12) describes the full batch setting, which can be expensive if the KG contains many edges. Therefore, in practice, whenever the graph is big, we adopt a stochastic evaluation of the REFACTOR layer by decomposing the evaluation into several mini-batches. We partition $\mathcal{T}$ into a set of computationally tractable mini-batches. For each of them, we restrict the neighbourhoods to the subgraph induced by it and readjust the computation of $n^l[v]$ to include only entities and edges present in it. We leave the investigation of other stochastic strategies (e.g. by taking Monte Carlo estimations of the expectations in Equation (11)) to future work. Finally, we cascade the mini-batch evaluation to produce one full layer evaluation.

**Training.** The learnable parameters of REFACTOR GNNs are the relation embeddings $\psi$, which parameterise $g(r)$ in $q^l_M, l \in [1, L]$. Inspired by [20, 21], we learn $\psi$ by layer-wise (stochastic) gradient descent. This is in contrast to conventional GNN training, where we need to backpropagate through all the layers. A (full-batch) GD training dynamic for $\psi$ can be written as $\psi_{t+1} = \psi_t - \eta \nabla \mathcal{L}_t(\psi_t)$, where $\mathcal{L}_t(\psi_t) = -|\mathcal{T}|^{-1} \sum_{\mathcal{T}} \log P_{\psi_t}(w|v,r)$, with:

$$P_{\psi_t}(w|v,r) = \text{Softmax}(\Gamma(v,r,\cdot))[w], \qquad \Gamma(v,r,w) = \langle h^t[v], h^t[w], g_{\psi_t}(r) \rangle$$

and the node state update as

$$h^t = \begin{cases} X & \text{if } t \bmod L = 0 \\ q^{t \bmod L}(\mathcal{T}, h^{t-1}) & \text{otherwise.} \end{cases} \qquad (13)$$

Implementation-wise, such a training dynamic equals to using an external memory for storing historical node states $h^{t-1}$ akin to the procedure introduced in [20]. The cached node states can then be queried to compute $h^t$ using Equation (12). From this perspective, we periodically clear *the node state cache* every $L$ full batches to force the model to predict based on on-the-fly $L$-layer message-passing. After training, we obtain $\psi^*$ and perform inference by running $L$-layer message-passing with

**Figure 1:** ReFactor GNN architecture – the left figure describes the messages (coloured edges) used to update the representation of node $v_1$, which depend on the type of relationship between the sender nodes and $v_1$ in the graph $G = \{(v_2, r_1, v_1), (v_3, r_2, v_1), (v_1, r_3, v_4)\}$; the right figure describes the computation graph for calculating $P(v \mid w, r)$, where $v, w \in \mathcal{E}$ and $r \in \mathcal{R}$: the embedding representations of $w$, $r$, and $v$ are used to score the edge $(w, r, v)$ via the scoring function $\Gamma$, which is then normalised via the Softmax function.

$\psi^*$. In general, $L$ determines the number of effective message-passing layers in REFACTOR GNNS. A larger $L$ enables REFACTOR GNNS to fuse information from more hops of neighbourhoods into the final node representations. In the meantime, it reduces the inductive applicability of REFACTOR GNNS due to over-smoothing and computational requirements. In the extreme case of $L = \infty$, *where we never clear the node state cache during training*, the final cached node states will be used for inference. Note that this latter inference regime is inherently transductive since there will be no cached states for new nodes.

## B   Experiments

We perform experiments to answer the following questions regarding REFACTOR GNNS:

- **Q1.** REFACTOR GNNS are derived from a message-passing reformulation of FMs: do they also inherit FMs' predictive accuracy in *transductive* KGC tasks? (Appendix B.1)

- **Q2.** REFACTOR GNNS "inductivise" FMs. Are they more statistically accurate than other GNN baselines in *inductive* KGC tasks? (Appendix B.2)

- **Q3.** The term $n[v]$ involves nodes that are not in the 1-hop neighbourhood. Is such *augmented message passing* [19] necessary for good KGC performance? (Appendix B.3)

For transductive experiments, we used three well-established KGC datasets: *UMLS* [22], *CoDEx-S* [23], and *FB15K237* [24]. For inductive experiments, we used the inductive KGC benchmarks introduced by GraIL [17], which include 12 pairs of knowledge graphs: (*FB15K237_vi*, *FB15K237_vi_ind*), (*WN18RR_vi*, *WN18RR_vi_ind*), and (*NELL_vi*, *NELL_vi_ind*), where $i \in [1, 2, 3, 4]$, and (*_vi*, *_vi_ind*) represents a pair of graphs with a shared relation vocabulary and non-overlapping entities. We follow the standard KGC evaluation protocol by fully ranking all the candidate entities and computing two metrics using the ranks of the ground-truth entities: Mean Reciprocal Ranking (MRR) and Hit Ratios at Top K (Hits@$K$) with $K \in [1, 3, 10]$. For the inductive KGC, we additionally consider the partial-ranking evaluation protocol used by GraIL for a fair comparison. Empirically, we find full ranking more difficult than partial ranking, and thus more suitable for reflecting the differences among models on GraIL datasets – we would like to call for future work on GraIL datasets to also adopt a full ranking protocol on these datasets.

Our *transductive* experiments used $L = \infty$, i.e. node states cache is never cleared, as we wanted to see if REFACTOR GNNS ($L = \infty$) can reach the performance of the FMs; on the other hand, in our *inductive* experiments, we used REFACTOR GNNS with $L \in \{1, 2, 3, 6, 9\}$, since we wanted to test their performances in inductive settings akin to standard GNNs. We used a hidden size of 768 for the node representations. All the models are trained using $[128, 512]$ in-batch negative samples and one global negative node for each positive link. We performed a grid search over the other hyper-parameters and selected the best configuration based on the validation MRR. Since training

---

[1]Typically until convergence, possibly by partially warm-starting $\theta$.

| Entity Encoder | UMLS | CoDEx-S | FB15K237 |
|---|---|---|---|
| R-GCN | - | 0.33 | 0.25 |
| Lookup (FM, specif. DistMult) | 0.90 | 0.43 | 0.30 |
| REFACTOR GNNS ($L = \infty$) | 0.93 | 0.44 | 0.33 |

**Table 1:** Test MRR for transductive KGC tasks.

deep GNNs with full-graph message passing might be slow for large knowledge graphs, we follow the literature [25–27] to sample sub-graphs for training GNNs. Considering that sampling on the fly often prevents high utilisation of GPUs, we resort to a two-stage process: we first sampled and serialised sub-graphs around the target edges in the mini-batches; we then trained the GNNs with the serialised sub-graphs. To ensure we have sufficient sub-graphs for training the models, we sampled for 20 epochs for each knowledge graph, i.e. 20 full passes over the full graph. The sub-graph sampler we currently used is LADIES [26].

## B.1 REFACTOR GNNS for Transductive Learning (Q1)

REFACTOR GNNS are derived from the message-passing reformulation of FMs. We expect them to approximate the performance of FMs for transductive KGC tasks. To verify this, we run experiments on the datasets UMLS, CoDEx-S, and FB15K237. For a fair comparison, we use Equation (2) as the decoder and consider i) lookup embedding table as the entity encoder, which forms the FM when combined with the decoder (Section 2.1), and ii) REFACTOR GNNS as the entity encoder. REFACTOR GNNS are trained with $L = \infty$, i.e. we never clear the node state cache. Since transductive KGC tasks do not involve new entities, the node state cache in REFACTOR GNNS can be directly used for link prediction. Table 1 summarises the result. We observe that REFACTOR GNNS achieve a similar or slightly better performance compared to the FM. This shows that REFACTOR GNNS are able to capture the essence of FMs and thus maintain strong at transductive KGC.

## B.2 REFACTOR GNNS for Inductive Learning (Q2)

Despite FMs' good empirical performance on transductive KGC tasks, they fail to be inductive as GNNs. According to our reformulation, this is due to the infinite message-passing layers hidden in FMs' optimisation. Discarding infinite message-passing layers, REFACTOR GNNS enable FMs to perform inductive reasoning tasks by learning to use a finite set of message-passing layers for prediction similarly to GNNs.

Here we present experiments to verify REFACTOR GNNS's capability for inductive reasoning. Specifically, we study the task of inductive KGC and investigate whether REFACTOR GNNS can generalise to unseen entities. Following [17], on GraIL datasets, we trained models on the original graph, and run 0-shot link prediction on the _ind test graph. Similar to the transductive experiments, we use Equation (2) as the decoder and vary the entity encoder. We denote three-layer REFACTOR GNNS as REFACTOR GNNS (3) and six-layer REFACTOR GNNS as REFACTOR GNNS (6). We consider several baseline entity encoders: i) no-pretrain, models without any pretraining on the original graph; ii) GAT(3), three-layer graph attention network [28]; iii) *GAT(6)*, six-layer graph attention network; iv) GraIL, a sub-graph-based relational GNN [17]; v) NBFNet, a path-based GNN [9], current SoTA on GraIL datasets. In addition to randomly initialised vectors as the node features, we also used textual node features, RoBERTa [29] Encodings of the entity descriptions, which are produced by SentenceBERT [30]. Due to space reason, we present the results on (*FB15K237_v*1, *FB15K237_v*1*_ind*) in Figure 2. Results on other datasets are similar and can be found in the appendix. We can see that without textual node features, REFACTOR GNNS perform better than GraIL (+23%); with textual node features, REFACTOR GNNS outperform both GraIL (+43%) and NBFNet (+10%), achieving new SoTA results on inductive KGC.

**Performance vs Parameter Efficiency as #Message-Passing Layers Increases.** Usually, as the number of message-passing layers increases in GNNs, the over-smoothing issue occurs while the computational cost also increases exponentially. REFACTOR GNNS avoid this by layer-wise training and sharing the weights across layers. Here we compare REFACTOR GNNS with $\{1, 3, 6, 9\}$ message-passing layer(s) with same-depth GATs – results are summarised in Figure 3. We observe

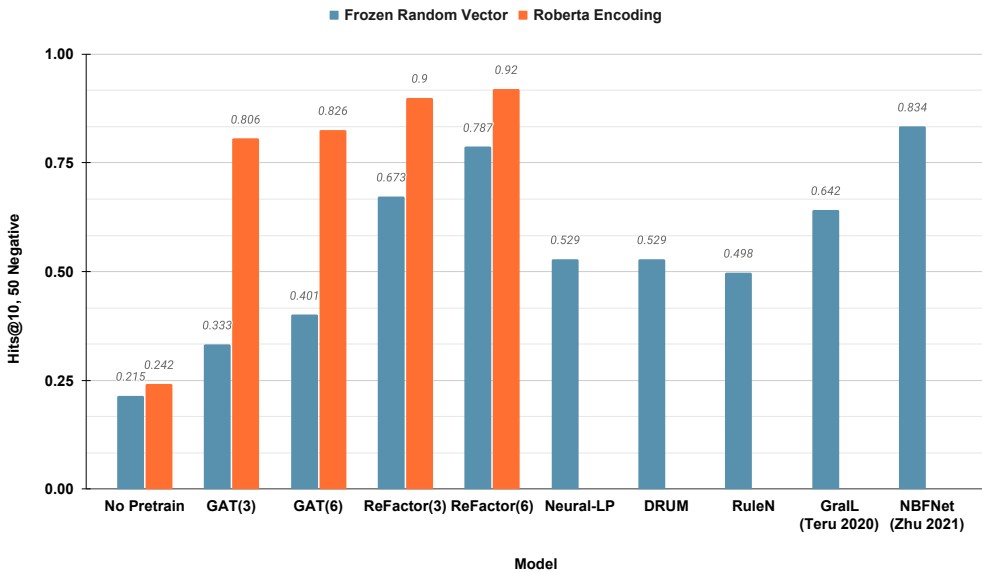

**Figure 2:** Inductive KGC Performance. Models are trained on the KG *FB15K237_v1* and tested on another KG *FB15K237_v1_ind*, where the entities are completely new. The results of GraIL and NBFNet are taken from [9]. The grey bars indicate methods that are not devised to incorporate node features.

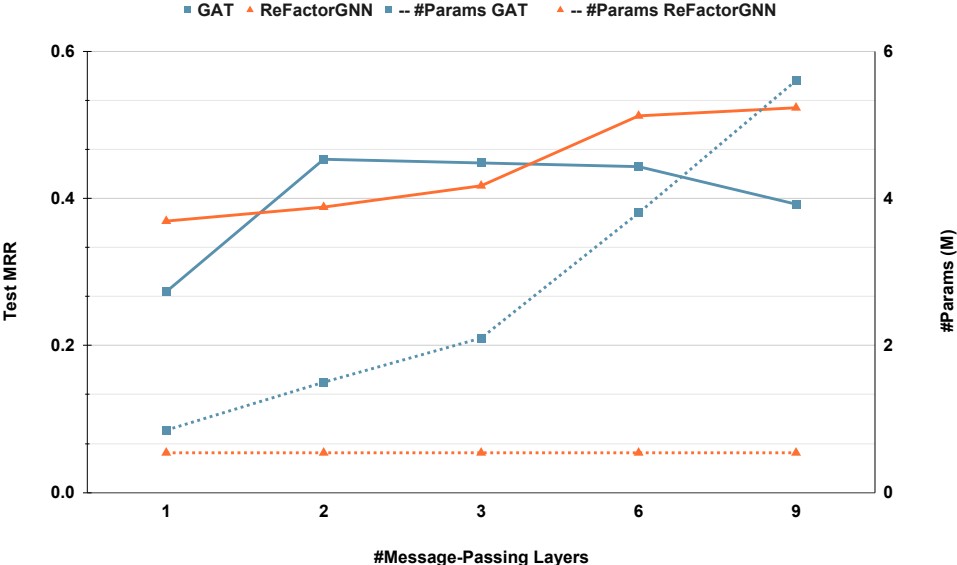

**Figure 3:** Performance vs Parameter Efficiency as #Layers Increases on *FB15K237_v1*. The left axis is Test MRR while the right axis is #Parameters. The solid lines and dashed lines indicate the changes of Test MRR and the changes of #Parameters.

| Test MRR | With Random Features | With Textual Features |
|---|---|---|
| with $n[v]$ | 0.425 | 0.486 |
| without $n[v]$ | 0.418 | 0.452 |

**Table 2:** Ablation on $n[v]$ for REFACTOR GNNS (6) trained on *FB15K237_v1*.

that increasing the number of message-passing layers in GATs does not necessarily improve the predictive accuracy – the best results were obtained with 3 message-passing layers on *FB15K237_v1* while using 6 and 9 layers leads to performance degradation. On the other hand, REFACTOR GNNS obtain consistent improvements when increasing #Layers from 1 to 3, 6, and 9. REFACTOR GNNS $(6, 6)$ and $(9, 9)$ clearly outperform their GAT counterparts. Most importantly, REFACTOR GNNS are more parameter-efficient than GATs, with a constant #Parameters as #Layers increases.

### B.3 Beyond Message-Passing (Q3)

As shown by Theorem 3.1, REFACTOR GNNS contain not only terms capturing information flow from the 1-hop neighbourhood, which falls into the classic message-passing framework, but also a term $n[v]$ that involve nodes outside the 1-hop neighbourhood. The term $n[v]$ can be treated as *augmented message-passing* on a dynamically rewired graph [19]. Here we perform ablation experiments to measure the impact of the $n[v]$ term. Table 2 summarises the ablation results: we can see that, without the term $n[v]$, REFACTOR GNNS with random vectors as node features yield a 2% lower MRR, while REFACTOR GNNS with RoBERTa textual encodings as node features produce a 7% lower MRR. This suggests that augmented message-passing also plays a significant role in REFACTOR GNNS' generalisation properties in downstream link prediction tasks. Future work might gain more insights by further dissecting the $n[v]$ term.

## C   Related Work

**Multi-Relational Graph Representation Learning.**   Multi-relational graph representation learning concerns graphs with various edge types. Another relevant line of work would be representation learning over heterogeneous graphs, where node types are also considered. Previous work on multi-relational graph representation learning focused either on FMs [3–5, 11, 12, 31–33] or on GNN-based models [16, 34–36]. Similar to a recent finding in a benchmark study over heterogeneous GNNs [37], where the best choices of GNNs for heterogeneous graphs seem to regress to simple homogeneous GNN baselines, the progress of multi-relational graph representation learning also mingles with FMs, the very classic multi-relational link predictors. Recently, FMs were found to be significantly more accurate than GNNs in KGC tasks, when coupled with specific training strategies [10, 11, 38]. While more advanced GNNs [9] for KBC are showing promise at the cost of extra algorithm complexity, little effort has been devoted to establishing the links between plain GNNs and FMs, which are strong multi-relational link predictors despite their simplicity. Our work aims to *align* GNNs with FMs so that we can combine the strengths from both families of models.

**Relationships between FMs and GNNs.**   A very recent work [39] builds a theoretical link between structural GNNs and node (positional) embeddings. However, on one end of the link, the second model category encompasses not merely factorisation-based models but also many practical graph neural networks, between which the connection is unknown. Our work instead offers a more practical link between positional node embeddings produced by FMs and positional node embeddings produced by GNNs, while at the same time focusing on KGC. Beyond FMs in KGC, using graph signal processing theory, [40] show that matrix factorisation (MF) based recommender models correspond to ideal low-pass graph convolutional filters. They also find infinite neighbourhood coverage in MF although using a completely different approach and focusing on a different domain in contrast to our work.

**Message Passing.**   Message-passing is itself a broad terminology – people generally talk about it under two different contexts. Firstly, as a computational technique, message passing allows recursively decomposing a global function into simple local, parallelisable computations [41], thus being widely used for solving inference problems in a graphical model. Specifically, we note that message passing-based inference techniques were proposed for matrix completion-based recommendation [42] and

Bayesian Boolean data decomposition [43] in the pre-deep-learning era. Secondly, as a paradigm of parameterising learnable functions over *graph-structured data*, message-passing has recently been used to provide a unified reformulation [2] for various GNN architectures, including Graph Attention Networks [28], Gated Graph Neural Networks [44], and Graph Convolutional Networks [1]. In this work, we show that FMs can also be cast as a special type of message-passing GNNs by considering the gradient descent updates [45] over node embeddings as message-passing operations between nodes. To the best of our knowledge, our work is the first to provide such connections between FMs and message-passing GNNs. We show that FMs can be seen as instances of GNNs, with a characteristic feature about the nodes being considered during the message-passing process: our REFACTOR GNNS can be seen as using an *Augmented Message-Passing* process on a dynamically re-wired graph [19].

## D  Conclusion & Future Work

**Conclusion.**    Motivated by the need of understanding FMs and GNNs despite the sharp differences in their analytical forms, our work establishes a link between FMs and GNNs on the task of multi-relational link prediction. The reformulation of FMs as GNNs addresses the question why FMs are stronger multi-relational link predictors compared to plain GNNs. Guided by the reformulation, we further propose a new variant of GNNs, REFACTOR GNNS, which combines the strengths of both GNNs and classic FMs. Empirical experiments show that REFACTOR GNNS produce significantly more accurate results than our GNN baselines on inductive link prediction tasks.

**Limitations.**    Since we adopted a two-stage (sub-graph serialisation and then model training) approach instead of online sampling, there can be side effects from the low sub-graph diversity. In our experiments, we used LADIES [26] for sub-graph sampling. Experiments with different sub-graph sampling algorithms, such as GraphSaint [27] might affect the downstream link prediction results. Furthermore, it would be interesting to analyse decoders other than DistMult, as well as additional optimisation schemes beyond SGD and AdaGrad. We do not dive deeper into the expressiveness of REFACTOR GNNS. Nevertheless, we offer a brief discussion in Appendix J.

**Future Work.**    The most direct future work would be using the insight to develop more sophisticated models at the crossroad between FMs and GNNs, e.g. by further parameterising the message/update function. One implication from our work is that reformulating FMs as message-passing enables the idea of "learning to factorize". This might broaden the usages of FMs, going beyond link prediction, to tasks such as graph classification. It could also be possible to connect REFACTOR GNNS with EP [46], another graph representation learning framework, since they share the same spirit of layer-wise backwards with an unsupervised loss. Another implication comes from our approach of unpacking embedding updates into a series of message-passing operations. This approach can be generalised to other dot-product-based models that use embedding layers for processing the inputs. In this sense, our work paves the way towards understanding complicated attention-based models e.g. transformers. Although transformers can be treated as GNNs over fully-connected graphs, where a sentence would be a graph and its tokens would be the nodes, the message-passing is limited to within each sentence under this view. We instead envision cross-sentence message-passing by reformulating the updates of the token embedding layer in transformers. In general, the direction of organising FMs, GNNs, and transformers under the same framework will allow a better understanding of all three models.

## E  Theorem 1 Proof

In this section, we prove Theorem 1, which we restate here for convenience.

**Theorem E.1** (Message passing in FMs)**.** *The gradient descent operator* GD *(7) on the node embeddings of a DistMult model (Equation* (4)*) with the maximum likelihood objective in Equation* (3) *and a multi-relational graph* $\mathcal{T}$ *defined over entities* $\mathcal{E}$ *induces a message-passing operator whose*

*composing functions are:*

$$q_{\text{M}}(\phi[v], r, \phi[w]) = \begin{cases} \phi[w] \odot g(r) & \text{if } (r, w) \in \mathcal{N}^1_+[v], \\ (1 - P_\theta(v|w, r))\phi[w] \odot g(r) & \text{if } (r, w) \in \mathcal{N}^1_-[v]; \end{cases} \tag{14}$$

$$q_{\text{A}}(\{m[v, r, w] : (r, w) \in \mathcal{N}^1[v]\}) = \sum_{(r, w) \in \mathcal{N}^1[v]} m[v, r, w]; \tag{15}$$

$$q_{\text{U}}(\phi[v], z[v]) = \phi[v] + \alpha z[v] - \beta n[v], \tag{16}$$

*where, defining the sets of triplets* $\mathcal{T}^{-v} = \{(s, r, o) \in \mathcal{T} : s \neq v \wedge o \neq v\}$,

$$n[v] = \frac{|\mathcal{N}^1_+[v]|}{|\mathcal{T}|} \mathbb{E}_{P_{\mathcal{N}^1_+[v]}} \mathbb{E}_{u \sim P_\theta(\cdot|v, r)} g(r) \odot \phi[u] + \frac{|\mathcal{T}^{-v}|}{|\mathcal{T}|} \mathbb{E}_{P_{\mathcal{T}^{-v}}} P_\theta(v|s, r) g(r) \odot \phi[s], \tag{17}$$

*where* $P_{\mathcal{N}^1_+[v]}$ *and* $P_{\mathcal{T}^{-v}}$ *are the empirical probability distributions associated to the respective sets.*

*Proof.* Remember that we assume that there are no triplets where the source and the target node are the same (i.e. $(v, r, v)$, with $v \in \mathcal{E}$ and $r \in \mathcal{R}$), and let $v \in \mathcal{E}$ be a node in $\mathcal{E}$. First, let us consider the gradient descent operator GD over $v$'s node embedding $\phi[v]$:

$$\text{GD}(\phi, \mathcal{T})[v] = \phi[v] + \alpha \sum_{(\bar{v}, \bar{r}, \bar{w}) \in \mathcal{T}} \frac{\partial \log P(\bar{w} \,|\, \bar{v}, \bar{r})}{\partial \phi[v]}.$$

The gradient is a sum over components associated with the triplets $(\bar{v}, \bar{r}, \bar{w}) \in \mathcal{T}$; based on whether the corresponding triplet involves $v$ in the subject or object position, or does not involve $v$ at all, these components can be grouped into three categories:

1. Components corresponding to the triplets where $\bar{v} = v \wedge \bar{w} \neq v$. The sum of these components is given by:

$$\sum_{(v, \bar{r}, \bar{w}) \in \mathcal{T}} \frac{\partial \log P(\bar{w} \,|v, \bar{r})}{\partial \phi[v]} = \sum_{(v, \bar{r}, \bar{w}) \in \mathcal{T}} \left[ \frac{\partial \Gamma(v, \bar{r}, \bar{w})}{\partial \phi[v]} - \sum_u P(u|v, \bar{r}) \frac{\partial \Gamma(v, \bar{r}, u)}{\partial \phi[v]} \right]$$

$$= \sum_{(\bar{r}, \bar{w}) \in \mathcal{N}^1_+[v]} \phi[\bar{w}] \odot g(\bar{r}) - \sum_{(v, \bar{r}, \bar{w}) \in \mathcal{T}} \sum_u P(u|v, \bar{r}) g(\bar{r}) \odot \phi[u].$$

2. Components corresponding to the triplets where $\bar{v} \neq v \wedge \bar{w} = v$. The sum of these components is given by:

$$\sum_{(\bar{v}, \bar{r}, v) \in \mathcal{T}} \frac{\partial \log P(v \,|\, \bar{v}, \bar{r})}{\partial \phi[v]} = \sum_{(\bar{v}, \bar{r}, v) \in \mathcal{T}} \left[ \frac{\partial \Gamma(\bar{v}, \bar{r}, v)}{\partial \phi[v]} - \sum_u P(u| \bar{v}, \bar{r}) \frac{\partial \Gamma(\bar{v}, \bar{r}, u)}{\partial \phi[v]} \right]$$

$$= \sum_{(\bar{v}, \bar{r}) \in \mathcal{N}^1_-[v]} g(\bar{r}) \odot \phi[\bar{v}] (1 - P(v| \bar{v}, \bar{r})).$$

3. Components corresponding to the triplets where $\bar{v} \neq v \wedge \bar{w} \neq v$. The sum of these components is given by:

$$\sum_{(\bar{v}, \bar{r}, \bar{w}) \in \mathcal{T}} \frac{\partial \log P(\bar{w} \,|\, \bar{v}, \bar{r})}{\partial \phi[v]} = \sum_{(\bar{v}, \bar{r}, \bar{w}) \in \mathcal{T}} \left[ 0 - \sum_u P(u| \bar{v}, \bar{r}) \frac{\partial \Gamma(\bar{v}, \bar{r}, u)}{\partial \phi[v]} \right]$$

$$= \sum_{(\bar{v}, \bar{r}, \bar{w}) \in \mathcal{T}} -P(v| \bar{v}, \bar{r}) \frac{\partial \Gamma(\bar{v}, \bar{r}, v)}{\partial \phi[v]}.$$

$$= \sum_{(\bar{v}, \bar{r}, \bar{w}) \in \mathcal{T}} -P(v| \bar{v}, \bar{r}) g(\bar{r}) \odot \phi[\bar{v}].$$

Collecting these three categories, the GD operator over $\phi[v]$, or rather the node representation update in DistMult, can be rewritten as:

$$\mathrm{GD}(\phi, \mathcal{T})[v] = \phi[v] + \alpha \underbrace{\sum_{\{(\bar{\mathrm{r}},\bar{\mathrm{w}}) \in \mathcal{N}_+^1[v]\}} \phi[\bar{\mathrm{w}}] \odot g(\bar{\mathrm{r}}) + \sum_{(\bar{\mathrm{r}},\bar{\mathrm{v}}) \in \mathcal{N}_-^1[v]} \phi[\bar{\mathrm{v}}] \odot g(\bar{\mathrm{r}})\left(1 - P(v \,|\, \bar{\mathrm{v}}, \bar{\mathrm{r}})\right)}_{v\text{'s neighbourhood} \to v} \quad (18)$$

$$-\alpha \underbrace{\sum_{(\bar{\mathrm{v}},\bar{\mathrm{r}},\bar{\mathrm{w}}) \in \mathcal{T}, \bar{\mathrm{v}} \neq v, \bar{\mathrm{w}} \neq v} P(v \,|\, \bar{\mathrm{v}}, \bar{\mathrm{r}}) g(\bar{\mathrm{r}}) \odot \phi[\bar{\mathrm{v}}] - \alpha \sum_{(v,\bar{\mathrm{r}},\bar{\mathrm{w}}) \in \mathcal{T}} \sum_u P(u|v, \bar{\mathrm{r}}) g(\bar{\mathrm{r}}) \odot \phi[u]}_{\text{beyond neighbourhood} \to v}. \quad (19)$$

Note that the component "$v$'s neighbourhood $\to v$" (highlighted in red) in Equation (18) is a sum over $v$'s neighbourhood – gathering information from positive neighbours $\phi[\bar{\mathrm{w}}], (\cdot, \bar{\mathrm{w}}) \in \mathcal{N}_+^1[v]$ and negative neighbours $\phi[\bar{\mathrm{v}}], (\cdot, \bar{\mathrm{v}}) \in \mathcal{N}_-^1[v]$. Hence, each atomic term of the sum can be seen as a message vector between $v$ and $v$'s neighbouring node. Formally, letting $w$ be $v$'s neighbouring node, the message vector can be written as follows

$$m[v, r, w] = q_{\mathrm{M}}(\phi[v], r, \phi[w]) = \begin{cases} \phi[w] \odot g(r), \text{ if } (r, w) \in \mathcal{N}_+^1[v], \\ \phi[w] \odot g(r)(1 - P(v|w, r)), \text{ if } (r, w) \in \mathcal{N}_-^1[v], \end{cases} \quad (20)$$

which induces a bi-directional message function $q_M$. On the other hand, the summation over these atomic terms (message vectors) induces the aggregate function $q_{\mathrm{A}}$:

$$z[v] = q_{\mathrm{A}}(\{m[v, r, w] \,:\, (r, w) \in \mathcal{N}^1[v]\})$$
$$= \sum_{(\bar{\mathrm{r}},\bar{\mathrm{w}}) \in \mathcal{N}_+^1[v]} m^l[v, \bar{\mathrm{r}}, \bar{\mathrm{w}}] + \sum_{(\bar{\mathrm{r}},\bar{\mathrm{v}}) \in \mathcal{N}_-^1[v]} m^l[\bar{\mathrm{v}}, \bar{\mathrm{r}}, v] = \sum_{(r, w) \in \mathcal{N}^1[v]} m[v, r, w]. \quad (21)$$

Finally, the component "beyond neighbourhood $\to v$" (highlighted in blue) is a term that contains dynamic information flow from global nodes to $v$. If we define:

$$n[v] = \frac{1}{|\mathcal{T}|} \sum_{(v,\bar{\mathrm{r}},\bar{\mathrm{w}}) \in \mathcal{T}} \sum_u P(u|v, \bar{\mathrm{r}}) g(\bar{\mathrm{r}}) \odot \phi[u] + \frac{1}{|\mathcal{T}|} \sum_{(\bar{\mathrm{v}},\bar{\mathrm{r}},\bar{\mathrm{w}}) \in \mathcal{T}, \bar{\mathrm{v}} \neq v, \bar{\mathrm{w}} \neq v} P(v \,|\, \bar{\mathrm{v}}, \bar{\mathrm{r}}) g(\bar{\mathrm{r}}) \odot \phi[\bar{\mathrm{v}}],$$

the GD operator over $\phi[v]$ then boils down to an update function which utilises previous node state $\phi[v]$, aggregated message $z[v]$ and a global term $n[v]$ to produce the new node state:

$$\mathrm{GD}(\phi, \mathcal{T})[v] = q_{\mathrm{U}}(\phi[v], z[v]) = \phi[v] + \alpha z[v] - \beta n[v]. \quad (22)$$

Furthermore, $n[v]$ can be seen as a weighted sum of expectations by recasting the summations over triplets as expectations:

$$n[v] = \frac{|\mathcal{N}_+^1[v]|}{|\mathcal{T}|} \mathbb{E}_{(v,\bar{\mathrm{r}},\bar{\mathrm{w}}) \sim P_{\mathcal{N}_+^1[v]}} \mathbb{E}_{u \sim P(\cdot|v,\bar{\mathrm{r}})} g(\bar{\mathrm{r}}) \odot \phi[u] + \frac{|\mathcal{T}^{-v}|}{|\mathcal{T}|} \mathbb{E}_{(\bar{\mathrm{v}},\bar{\mathrm{r}},\bar{\mathrm{w}}) \sim P_{\mathcal{T}^{-v}}} P(v \,|\, \bar{\mathrm{v}}, \bar{\mathrm{r}}, ) g(\bar{\mathrm{r}}) \odot \phi[\bar{\mathrm{v}}] \quad (23)$$

where $\mathcal{T}^{-v} = \{(\bar{\mathrm{v}}, \bar{\mathrm{r}}, \bar{\mathrm{v}}') \in \mathcal{T} \,|\, \bar{\mathrm{v}} \neq v \wedge \bar{\mathrm{v}}' \neq v\}$ is the set of triplets that do not contain $v$. $\qquad \square$

## E.1 Extension to AdaGrad and N3 Regularisation

State-of-the-art FMs are often trained with training strategies adapted for each model category. For example, using an N3 regularizer [11] and AdaGrad optimiser [47], which we use for our experiments. For N3 regularizer, we add a gradient term induced by the regularised loss:

$$\frac{\partial L}{\partial \phi[v]} = \frac{\partial L_{\mathrm{fit}}}{\partial \phi[v]} + \lambda \frac{\partial L_{\mathrm{reg}}}{\partial \phi[v]} = \frac{\partial L_{\mathrm{fit}}}{\partial \phi[v]} + \lambda \mathrm{sign}(\phi[v]) \phi[v]^2$$

where $L_{\mathrm{fit}}$ is the training loss, $L_{\mathrm{reg}}$ is the regularisation term, $\mathrm{sign}(\cdot)$ is a element-wise sign function, and $\lambda \in \mathbb{R}_+$ is a hyper-parameter specifying the regularisation strength. The added component relative to this regularizer fits into the message function as follows:

$$q_{\mathrm{M}}(\phi[v], r, \phi[w]) = \begin{cases} \phi[w] \odot g(r) - \lambda \mathrm{sign}(\phi[w]) \phi[w]^2, \text{ if } (r, w) \in \mathcal{N}_+^1[v], \\ \phi[w] \odot g(r)(1 - P(v|w, r)) - \lambda \mathrm{sign}(\phi[w]) \phi[w]^2, \text{ if } (w, r) \in \mathcal{N}_-^1[v]; \end{cases} \quad (24)$$

Our derivation in Section 3 focuses on (stochastic) gradient descent as the optimiser for training FMs. Going beyond this, complex gradient-based optimisers like AdaGrad use running statistics of the gradients. For example, for an AdaGrad optimiser, the gradient is element-wisely re-scaled by $\frac{1}{\sqrt{s_v}+\epsilon}\nabla_{\phi[v]}L$ where $s$ is the running sum of squared gradients and $\epsilon > 0$ is a hyper-parameter added to the denominator to improve numerical stability. Such re-scaling can be absorbed into the update equation:

$$\text{AdaGrad}(\phi, \mathcal{T})[v] = \phi[v] + (\alpha z[v] - \beta n[v]) * \frac{1}{\sqrt{s[v]} + \epsilon}.$$

In general, we can interpret any auxiliary variable introduced by the optimizer (e.g. the velocity) as an additional part of the entities and relations representations on which message passing happens. However, the specific equations would depend on the optimizer's dynamics and would be hard to formally generalise.

### E.2 Extensions to Other Score Functions e.g. ComplEx

The two main design choices in Theorem E.1 are 1) the score function $\Gamma$, and 2) the optimization dynamics over the node embeddings. In the paper, we chose DistMult and GD because of their mathematical simplicity, leading to easier-to-read formulas. We can adapt the theorem to general, smooth scoring functions $\Gamma : \mathcal{E} \times \mathcal{R} \times \mathcal{E} \to \mathbf{R}$ by replacing occurrences of the gradient of DistMult with a generic $\nabla\Gamma$ (the gradient of DistMult w.r.t. $\phi[v]$ at $(v, r, w)$ is simply $g(r) \odot \phi[w]$). This gives us the following lemma:

**Lemma E.2** (Message passing in FMs). *The gradient descent operator* GD *(7) on the node embeddings of a general score function with the maximum likelihood objective in Equation* (3) *and a multi-relational graph $\mathcal{T}$ defined over entities $\mathcal{E}$ induces a message-passing operator whose composing functions are:*

$$q_{\text{M}}(\phi[v], r, \phi[w]) = \begin{cases} \nabla_{\phi[v]}\Gamma(v, r, w) & \text{if } (r, w) \in \mathcal{N}_+^1[v], \\ (1 - P_\theta(v|w, r))\nabla_{\phi[v]}\Gamma(w, r, v) & \text{if } (r, w) \in \mathcal{N}_-^1[v]; \end{cases} \tag{25}$$

$$q_{\text{A}}(\{m[v, r, w] : (r, w) \in \mathcal{N}^1[v]\}) = \sum_{(r,w)\in\mathcal{N}^1[v]} m[v, r, w]; \tag{26}$$

$$q_{\text{U}}(\phi[v], z[v]) = \phi[v] + \alpha z[v] - \beta n[v], \tag{27}$$

*where, defining the sets of triplets $\mathcal{T}^{-v} = \{(s, r, o) \in \mathcal{T} : s \neq v \wedge o \neq v\}$,*

$$n[v] = \frac{|\mathcal{N}_+^1[v]|}{|\mathcal{T}|}\mathbb{E}_{P_{\mathcal{N}_+^1[v]}}\mathbb{E}_{u\sim P_\theta(\cdot|v,r)}\nabla_{\phi[v]}\Gamma(v, r, u) + \frac{|\mathcal{T}^{-v}|}{|\mathcal{T}|}\mathbb{E}_{P_{\mathcal{T}^{-v}}}P_\theta(v|s, r)\nabla_{\phi[v]}\Gamma(s, r, v), \tag{28}$$

*where $P_{\mathcal{N}_+^1[v]}$ and $P_{\mathcal{T}^{-v}}$ are the empirical probability distributions associated to the respective sets.*

Accordingly, the node representation updating equations in Section 3.1 can be re-written as follows

$$\text{GD}(\phi, \{(v, r, w)\})[v] = \phi[v] + \alpha\left(\underbrace{\nabla_{\phi[v]}\Gamma(v, r, w)}_{w\to v} - \underbrace{\sum_{u\in\mathcal{E}}P_\theta(u|v, r)\nabla_{\phi[v]}\Gamma(v, r, u)}_{u\to v}\right),$$

$$\text{GD}(\phi, \{(v, r, w)\})[w] = \phi[w] + \alpha\underbrace{(1 - P_\theta(w|v, r))\nabla_{\phi[w]}\Gamma(v, r, w)}_{v\to w},$$

$$\text{GD}(\phi, \{(v, r, w)\})[u] = \phi[u] + \alpha\left(\underbrace{-P_\theta(u|v, r)\nabla_{\phi[u]}\Gamma(v, r, u)}_{v\to u}\right).$$

$\nabla_{\phi[\cdot]}\Gamma$ can be different for different models. For example, here we offer a specific derivation for ComplEx [5]. Let $d = K/2$ be the hidden size for ComplEx. The ComplEx score function is given as follows

$$\begin{aligned}
\Gamma(v,r,w) = &< \psi[r]_{(0:d)}, \phi[v]_{(0:d)}, \phi[w]_{(0:d)} > + < \psi[r]_{(0:d)}, \phi[v]_{(d:)}, \phi[w]_{(d:)} > \\
&+ < \psi[r]_{(d:)}, \phi[v]_{(0:d)}, \phi[w]_{(d:)} > - < \psi[r]_{(d:)}, \phi[v]_{(d:)}, \phi[w]_{(0:d)} >
\end{aligned} \tag{29}$$

where $(0:d)$ indicates the real part of the complex vector and $(d:)$ indicates the image part of the complex vector. The gradients of the ComplEx score function with respect to the real/image node representations are given by $\frac{\partial \Gamma(v,r,w)}{\partial \phi[v]_{(0:d)}} = \psi[r]_{(0:d)} \odot \phi[w]_{(0:d)} + \psi[r]_{(d:)} \odot \phi[w]_{(d:)}, \frac{\partial \Gamma(v,r,w)}{\partial \phi[v]_{(d:)}} = \psi[r]_{(0:d)} \odot \phi[w]_{(d:)} - \psi[r]_{(d:)} \odot \phi[w]_{(0:d)}, \frac{\partial \Gamma(v,r,w)}{\partial \phi[w]_{(0:d)}} = \psi[r]_{(0:d)} \odot \phi[v]_{(0:d)} - \psi[r]_{(d:)} \odot \phi[v]_{(d:)}, \frac{\partial \Gamma(v,r,w)}{\partial \phi[w]_{(d:)}} = \psi[r]_{(0:d)} \odot \phi[v]_{(d:)} + \psi[r]_{(d:)} \odot \phi[v]_{(0:d)}$. Concatenating gradients for the real part and the image part, we have the gradients

$$\nabla_{\phi[v]}\Gamma(v,r,w) = \frac{\partial \Gamma(v,r,w)}{\partial \phi[v]_{(0:d)}} \| \frac{\partial \Gamma(v,r,w)}{\partial \phi[v]_{(d:)}},$$

$$\nabla_{\phi[w]}\Gamma(v,r,w) = \frac{\partial \Gamma(v,r,w)}{\partial \phi[w]_{(0:d)}} \| \frac{\partial \Gamma(v,r,w)}{\partial \phi[w]_{(d:)}}.$$

## F  Additional Results on Inductive KGC Tasks

In this paper, we describe the results on FB15K237_v1_ind under some random seed. To confirm the significance and sensitivity, we further experiment with additional 5 random seeds. Due to our computational budget, for this experiment, we resorted to a coarse grid when performing the hyper-parameters sweeps. Following standard evaluation protocols, we report the mean values and standard deviations of the filtered Hits@10 over 5 random seeds. Numbers for Neural-LP, DRUM, RuleN, GraIL, and NBFNet are taken from the literature [9, 17]. "-" means the numbers are not applicable. Table 4 summarises the results. REFACTOR GNNS are able to make use of both types of input features, while textual features benefit both GAT and REFACTOR GNNS for most datasets. Increasing depth benefits WN18RR_v$i$_ind ($i \in [1,2,3,4]$) most. Future work could consider the impact of textual node features provided by different pre-trained language models. Another interesting direction is to investigate the impact of depth on GNNs for datasets like WN18RR, where many kinds of hierarchies are observed in the data.

In addition to the *partial ranking* evaluation protocol, where the ground-truth subject/object entity is ranked against 50 sampled entities,[2] we also consider the *full ranking* evaluation protocol, where the ground-truth subject/object entity is ranked against all the entities. Table 5 summarises the results. Empirically, we observe that *full ranking* is more suitable for reflecting the differences between models than *partial ranking*. It also has less variance than *partial ranking*, since it requires no sampling from the candidate entities. Hence, we believe there is good reason to recommend the community to use *full ranking* for these datasets in the future.

## G  Additional Results on The Impact of Meaningful Node Features

To better understand the impact that meaningful node features have on REFACTOR GNNS for the task of knowledge graph completion, we compare REFACTOR GNNS trained with RoBERTa Encodings (one example of meaningful node features) and REFACTOR GNNS trained with Random Vectors (not meaningful node features). We perform experiments on *FB15K237_v1* and vary the number of message-passing layers: $L \in \{3,6,\infty\}$. Table 3 summarises the differences. We can see that meaningful node features are highly beneficial if REFACTOR GNNS are only provided with a small number of message-passing layers. As more message-passing layers are allowed, the benefit of REFACTOR GNNS diminishes. The extreme case would be $L = \infty$, where the benefit of meaningful node features becomes negligible. We hypothesise that this might be why meaningful node features haven not been found to be useful for transductive knowledge graph completion.

---

[2]One implementation for such evaluation can be found in GraIL's codebase.

| Depth | 3 | 6 | $\infty$ |
|---|---|---|---|
| $\Delta$ Test MRR | 0.060 | 0.045 | 0.016 |

**Table 3:** The Impact of Meaningful Node Feature on *FB15K237_v1*. $\Delta$ Test MRR is computed by `test mrr (textual node features)` − `test mrr (random node features)`. Larger $\Delta$ means meaningful node features bring more benefit.

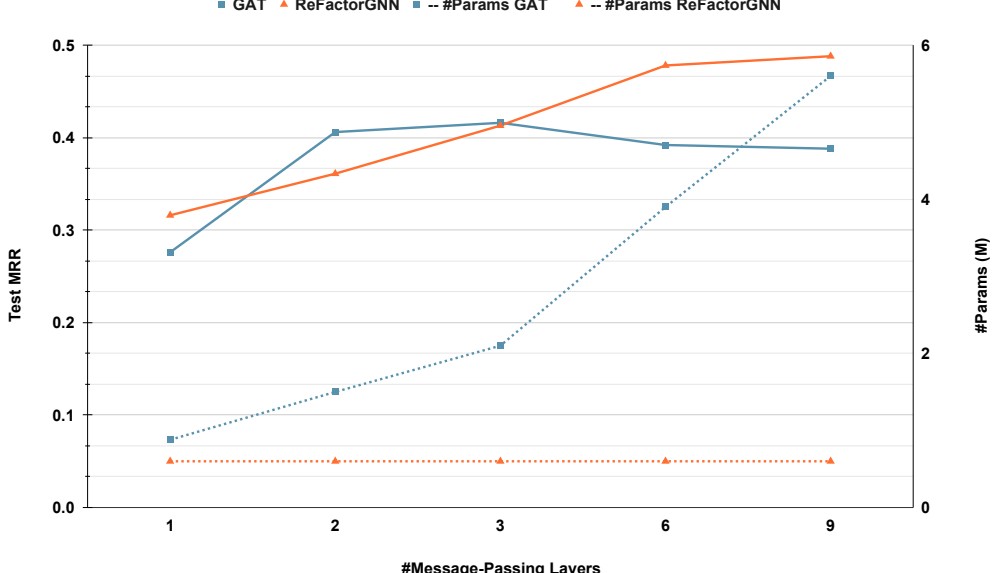

**Figure 4:** Performance vs Parameter Efficiency as #Layers Increases on *FB15K237_v2*. The left axis is Test MRR while the right axis is #Parameters. The solid lines and dashed lines indicate the changes of Test MRR and the changes of #Parameters.

**Table 4:** Hits@10 with Partial Ranking against 50 Negative Samples. "[T]" indicates using textual encodings of entity descriptions [30] as input (positional) node features; "[R]" indicates using frozen random vectors as input (positional) node feature.

| | WN18RR | | | | FB15k-237 | | | | NELL-995 | | | |
|---|---|---|---|---|---|---|---|---|---|---|---|---|
| | v1 | v2 | v3 | v4 | v1 | v2 | v3 | v4 | v1 | v2 | v3 | v4 |
| No Pretrain [R] | 0.220±0.048 | 0.226±0.013 | 0.244±0.020 | 0.218±0.050 | 0.215±0.019 | 0.207±0.008 | 0.211±0.002 | 0.205±0.008 | 0.543±0.022 | 0.207±0.008 | 0.216±0.004 | 0.198±0.006 |
| No Pretrain [T] | 0.267±0.020 | 0.236±0.020 | 0.292±0.025 | 0.253±0.022 | 0.242±0.018 | 0.227±0.007 | 0.240±0.011 | 0.244±0.003 | 0.538±0.079 | 0.234±0.017 | 0.242±0.020 | 0.191±0.036 |
| Neural-LP | 0.744 | 0.689 | 0.462 | 0.671 | 0.529 | 0.589 | 0.529 | 0.559 | 0.408 | 0.787 | 0.827 | 0.806 |
| DRUM | 0.744 | 0.689 | 0.462 | 0.671 | 0.529 | 0.587 | 0.529 | 0.559 | 0.194 | 0.786 | 0.827 | 0.806 |
| RuleN | 0.809 | 0.782 | 0.534 | 0.716 | 0.498 | 0.778 | 0.877 | 0.856 | 0.535 | 0.818 | 0.773 | 0.614 |
| GAT(3) [R] | 0.583±0.022 | 0.797±0.002 | 0.560±0.005 | 0.660±0.015 | 0.333±0.042 | 0.312±0.036 | 0.407±0.072 | 0.363±0.050 | 0.906±0.004 | 0.303±0.031 | 0.351±0.009 | 0.187±0.098 |
| GAT(6) [R] | 0.850±0.014 | 0.841±0.001 | 0.631±0.020 | 0.802±0.004 | 0.401±0.020 | 0.445±0.018 | 0.461±0.048 | 0.406±0.143 | 0.811±0.039 | 0.670±0.055 | 0.341±0.042 | 0.301±0.002 |
| GAT(3) [T] | **0.970±0.002** | 0.980±0.001 | 0.897±0.005 | 0.960±0.001 | 0.806±0.003 | 0.942±0.001 | 0.941±0.002 | 0.954±0.001 | 0.938±0.005 | 0.839±0.001 | 0.962±0.001 | 0.354±0.002 |
| GAT(6) [T] | 0.965±0.002 | 0.986±0.001 | 0.920±0.002 | 0.970±0.003 | 0.826±0.004 | 0.943±0.001 | 0.927±0.003 | 0.927±0.001 | 0.904±0.000 | 0.811±0.001 | 0.880±0.001 | 0.297±0.003 |
| GraIL | 0.825 | 0.787 | 0.584 | 0.734 | 0.642 | 0.818 | 0.828 | 0.893 | 0.595 | 0.933 | 0.914 | 0.732 |
| NBFNet | 0.948 | 0.905 | 0.893 | 0.890 | 0.834 | 0.949 | 0.951 | 0.960 | - | - | - | - |
| ReFactorGNN(3) [R] | 0.899±0.003 | 0.842±0.004 | 0.605±0.000 | 0.801±0.002 | 0.673±0.002 | 0.812±0.002 | 0.833±0.003 | 0.877±0.002 | 0.913±0.000 | 0.913±0.011 | 0.893±0.000 | 0.838±0.002 |
| ReFactorGNN(6) [R] | 0.885±0.000 | 0.854±0.003 | 0.738±0.006 | 0.817±0.004 | 0.787±0.004 | 0.903±0.003 | 0.903±0.002 | 0.920±0.002 | 0.971±0.007 | 0.957±0.003 | 0.935±0.003 | 0.927±0.001 |
| ReFactorGNN(3) [T] | 0.918±0.002 | 0.973±0.001 | 0.910±0.003 | 0.934±0.001 | 0.900±0.004 | 0.959±0.001 | 0.952±0.002 | 0.968±0.001 | **0.955±0.004** | 0.931±0.001 | 0.978±0.001 | 0.929±0.001 |
| ReFactorGNN(6) [T] | **0.970±0.002** | **0.988±0.001** | **0.944±0.002** | **0.987±0.000** | **0.920±0.001** | **0.963±0.001** | **0.962±0.002** | **0.970±0.002** | 0.949±0.011 | **0.963±0.001** | **0.994±0.000** | **0.955±0.002** |

**Table 5:** Hits@10 with Full Ranking against All Candidate Entities. "[T]" indicates using textual encodings of entity descriptions [30] as input (positional) node features; "[R]" indicates using frozen random vectors as input (positional) node feature.

| | WN18RR | | | | FB15k-237 | | | | NELL-995 | | | |
|---|---|---|---|---|---|---|---|---|---|---|---|---|
| | v1 | v2 | v3 | v4 | v1 | v2 | v3 | v4 | v1 | v2 | v3 | v4 |
| No Pretrain [R] | 0.020±0.006 | 0.004±0.001 | 0.004±0.003 | 0.003±0.001 | 0.013±0.003 | 0.012±0.001 | 0.004±0.001 | 0.002±0.001 | 0.255±0.021 | 0.004±0.001 | 0.001±0.001 | 0.003±0.001 |
| No Pretrain [T] | 0.027±0.009 | 0.007±0.003 | 0.006±0.001 | 0.005±0.001 | 0.014±0.001 | 0.010±0.001 | 0.007±0.001 | 0.006±0.001 | 0.262±0.031 | 0.006±0.002 | 0.006±0.002 | 0.003±0.001 |
| GAT(3) [R] | 0.171±0.008 | 0.504±0.026 | 0.260±0.022 | 0.089±0.017 | 0.074±0.003 | 0.050±0.014 | 0.051±0.019 | 0.023±0.012 | 0.806±0.019 | 0.003±0.002 | 0.008±0.007 | 0.008±0.004 |
| GAT(6) [R] | 0.575±0.005 | 0.698±0.003 | 0.312±0.000 | 0.606±0.002 | 0.048±0.004 | 0.028±0.004 | 0.033±0.018 | 0.015±0.026 | 0.491±0.112 | 0.110±0.048 | 0.031±0.010 | 0.031±0.002 |
| GAT(3) [T] | 0.794±0.000 | 0.826±0.000 | 0.468±0.000 | 0.705±0.000 | 0.331±0.000 | 0.585±0.000 | 0.505±0.000 | 0.449±0.000 | 0.856±0.000 | 0.245±0.000 | 0.345±0.000 | 0.078±0.000 |
| GAT(6) [T] | 0.815±0.000 | 0.808±0.000 | 0.469±0.000 | 0.701±0.000 | 0.416±0.000 | 0.483±0.000 | 0.391±0.000 | 0.388±0.000 | 0.851±0.000 | 0.189±0.000 | 0.137±0.000 | 0.023±0.000 |
| ReFactorGNN(3) [R] | 0.826±0.000 | 0.758±0.002 | 0.374±0.004 | 0.707±0.000 | 0.455±0.010 | 0.603±0.008 | 0.556±0.003 | 0.587±0.003 | 0.907±0.004 | 0.700±0.001 | 0.630±0.001 | 0.511±0.001 |
| ReFactorGNN(6) [R] | 0.826±0.001 | 0.769±0.005 | 0.440±0.001 | 0.731±0.000 | 0.558±0.007 | 0.694±0.006 | 0.639±0.006 | 0.640±0.000 | **0.967±0.005** | **0.764±0.009** | 0.697±0.005 | **0.703±0.001** |
| ReFactorGNN(3) [T] | 0.805±0.000 | 0.796±0.003 | 0.483±0.000 | 0.682±0.000 | 0.589±0.001 | 0.672±0.001 | 0.610±0.001 | 0.611±0.000 | 0.918±0.000 | 0.629±0.001 | 0.634±0.000 | 0.305±0.000 |
| ReFactorGNN(6) [T] | **0.844±0.004** | **0.848±0.003** | **0.522±0.001** | **0.781±0.001** | **0.619±0.000** | **0.721±0.001** | **0.663±0.000** | **0.660±0.000** | 0.913±0.000 | 0.733±0.000 | **0.711±0.000** | 0.417±0.000 |

## H    Additional Results on Parameter Efficiency

Figure 4 shows the parameter efficiency on the dataset *FB15K237_v2*.

## I    Discussion on Complexity

We can analyse the scalability of REFACTOR GNNS along three axes, the number of layers $L$, the embedding size $d$, and the number of triplets/edges in the graph $|\mathcal{T}|$. For scalability w.r.t. to the number of layers, let $L$ denote the number of message-passing layers. Since REFACTOR GNNS tie the weights across the layers, the parameter complexity of REFACTOR GNNS is $\mathcal{O}(1)$, while it is $\mathcal{O}(L)$ for standard GNNs such as GATs, GCNs, and R-GCNs. Additionally, since REFACTOR GNNS adopt layer-wise training enabled via the external memory for node state caching, the training memory footprint is also $\mathcal{O}(1)$ as opposed to $\mathcal{O}(L)$ for standard GNNs. For scalability w.r.t the embedding size, let $d$ denote the embedding size. REFACTOR GNNS scale linearly with $d$, as opposed to most GNNs in literature where the parameter and time complexities scale quadratically with $d$. For scalability w.r.t. the number of triplets/edges in the graph, we denote the entity set as $\mathcal{E}$, the relation set as $\mathcal{R}$, and the triplets as $\mathcal{T}$. NBFNet requires $O(LT^2d + LTVd^2)$ inference run-time complexity since the message-passing is done for every source node and query relation – quadratic w.r.t the number of triplets $\mathcal{T}$ while REFACTOR GNNS are of linear complexity w.r.t $\mathcal{T}$. Extending the complexity analysis in NBFNet [9] to all the triplets, we include a detailed table for complexity comparison in Table 6. The inference complexity refers to the cost per forward pass over the entire graph.

| | Parameter Complexity | Training Memory Complexity | Inference Memory Complexity | Training Time Complexity | Inference Time Complexity |
|---|---|---|---|---|---|
| GAT | $O(Ld^2)$ | $O(L|V|d)$ | $O(L|V|d)$ | $O(L|V|d^2 + L|T|d)$ | $O(L|V|d^2 + L|T|d)$ |
| R-GCN | $O(L|R|d^2)$ | $O(L|V|d)$ | $O(L|V|d)$ | $O(L|T|d^2 + L|V|d^2)$ | $O(L|T|d^2 + L|V|d^2)$ |
| NBFNet | $O(L|R|d^2)$ | $O(L|V||T|d)$ | $O(L|V||T|d)$ | $O(L|T|^2d + L|T||V|d^2)$ | $O(L|T|^2d + L|T||V|d^2)$ |
| REFACTOR GNNS | $O(|R|d)$ | $O(|V|d)$ | $O(L|V|d)$ | $O(|T||V|d)$ | $O(L|T||V|d)$ |

**Table 6:** Complexity Comparison.

## J    Discussion on Expressiveness of FMs, GNNs and REFACTOR GNNS

We envision one interesting branch of future work would be a unified framework of expressiveness for all three model categories: FMs, GNNs and REFACTOR GNNS. To the best of our knowledge, there are currently two separate notions of expressiveness, one for FMs and the other for GNNs. While these two notions of expressiveness are both widely acclaimed within their own communities, it is unclear how to bridge them and produce a new tool supporting the analysis of the empirical applications (REFACTOR GNNS) that seam the two communities.

**Fully Expressiveness for Adjacency Recovery.**    In the FM community, a FM is said to be *fully expressive* [48] if, for any given graph $\mathcal{T}$ over entities $\mathcal{E}$ and relations $\mathcal{R}$, it can fully reconstruct the input adjacency tensor with a embedding size bounded by $\min(|\mathcal{E}||\mathcal{R}|, |\mathcal{T}| + 1)$. We can generalise this expressiveness analysis to the spectrum of FM-GNN models (REFACTOR GNNS). In the $L \to \infty$ limit, REFACTOR GNNS are as fully expressive as the underlying FMs. In fact, a REFACTOR GNN based on DistMult [4] is not fully expressive (because of its symmetry); however a REFACTOR GNN based, e.g. on ComplEx [5, 11] can reach full expressiveness for $L \to \infty$.

**Weisfeiler-Leman Tests for Nodes/Graphs Separation.**    For GNNs, established results concern the separation power of induced representations in terms of Weisfeiler-Leman (WL) isomorphism tests [49, 50]. However, none of these results is directly applicable to our setting (e.g. they only consider one relationship). Nevertheless, if we consider our REFACTOR GNNS in a one-relationship, simple graph setting, following the formalism of [50], we note that the REFACTOR Layer function cannot be written in Guarded Tensor Language since at each layer it computes a global term $n[v]$. Moreover, REFACTOR GNNS only process information coming from two nodes at one time. These two facts imply that REFACTOR GNNS have a separation power upper bound comparable to the 1-WL test, i.e. comparable to 1-MPNN (not guarded). We note that it could also be meaningful to analyse the global term $n[v]$ in a similar fashion as [51] did with the global readout function.

We are not aware of explicit connections between the two above notions of expressiveness. We think there is some possibility that we can bridge them, which itself will be a very interesting research direction, but would require a very substantial amount of additional work and presentation space and is thus beyond the scope of this paper.

Alternatively, we can also increase the capacity REFACTOR GNNS by adding more parameters to the message, aggregation and update operators. For example, introducing additional MLPs to transform the input node features or include non-linearity in the GNN update operator. This would bring REFACTOR GNNS closer to architectures like KE-GCN [52], where the message functions still capture the nice properties of factorisation-based models but the composition of message, aggregate and update function might have impacts over whether these properties can be retained or not.

Another method for increasing expressive power for link prediction task only is to extend ReFactor GNNs from node-wise to pairwise (Sec 2.2 in our paper) representations like GraIL [17] and NBFNet [9], which is more computationally intensive, but yields more powerful as node representations are not standalone but adapted to a specific query.

## K   Experimental Details: Setup, Hyper-Parameters, and Implementation

As we stated in the experiments section, we used a two-stage training process. In stage one, we sample subgraphs around query links and serialise them. In stage two, we load the serialised subgraphs and train the GNNs. For transductive knowledge graph completion, we test the model on the same graph (but different splits). For inductive knowledge graph completion, we test the model on the new graph, where the relation vocabulary is shared with the training graph, while the entities are novel. We use the validation split for selecting the best hyper-parameter configuration and report the corresponding test performance. We include reciprocal triplets into the training triplets following standard practice [11].

For subgraph serialisation, we first sample a mini-batch of triplets and then use these nodes as seed nodes for sampling subgraphs. We also randomly draw a node globally and add it to the seed nodes. The training batch size is 256 while the valid/test batch size is 8. We use the LADIES algorithm [26] and sample subgraphs with depths in $[1, 2, 3, 6, 9]$ and a width of 256. For each graph, we keep sampling for 20 epochs, i.e. roughly 20 full passes over the graph.

For general model training, we consider hyper-parameters including learning rates in $[0.01, 0.001]$, weight decay values in $[0, 0.1, 0.01]$, and dropout values in $[0, 0.5]$. For GATs, we use 768 as the hidden size and 8 as the number of attention heads. We train GATs with 3 layers and 6 layers. We also consider whether or not to combine the outputs from all the layers. For REFACTOR GNNS, we use the same hidden size as GAT. We consider whether the ReFactor Layer is induced by a SGD operator or by a AdaGrad operator. Within a ReFactor Layer, we also consider the N3 regulariser strength values $[0, 0.005, 0.0005]$, the $\alpha$ values $[0.1, 0.01]$, and the option of removing the $n[v]$, where the message-passing layer only involves information flow within 1-hop neighbourhood as most the classic message-passing GNNs do.

We use grid search to find the best hyper-parameter configuration based on the validation MRR. Each training run is done using two Tesla V100 (16GB) GPUs with, where data parallelism was implemented via the *DistributedDataParallel* component of *pytorch-lightning*. For inductive learning experiments, inference for all the validation and test queries on small datasets like FB15K237_v1 takes about 1-5 seconds, while on medium datasets it takes approximately 20 seconds, and on big datasets like WN18RR_v4 it requires approximately 60 seconds. For most training runs, the memory footprint is less than 40% (13GB). The training time for 20 full passes over the graph is about 1, 7, and 21 minutes respectively for small, medium, and large datasets.

Our code will be available at ReFactorGNN in recent future. Ping us with email if you need an urgent copy. We adapted the LADIES subgraph sampler from the GPT-GNN codebase [53] for sampling on knowledge graphs. The datasets we used can be downloaded from the repositories Datasets for Knowledge Graph Completion with Textual Information about Entities and GraIL - Graph Inductive Learning. We implemented REFACTOR GNNS using the *MessagePassing* API in *PyTorch Geometric*. Specially, we used *message_and_aggregate* function to compute the aggregated messages.

