# OpenReview forum: "ReFactorGNNs: Revisiting Factorisation-based Models from a Message-Passing Perspective"
_logconference.io/LOG/2022/Conference — LoG 2022 Poster_

### Official Review · Reviewer_2e63 · 2022-10-07

**Overall Score:** 8
**Confidence:** 5

**Review:**

### Summary ###
This extended abstract proposes a new approach, ReFACTOR GNN, that lies at the intersection of factorisation-based matrix factorisation methods and standard GNNs. In particular, the work revisits factorisation-based methods, such as DistMult, from the lens of gradient descent (GD), and shows that GD steps can be mapped to a message passing iteration in a GNN. More specifically, the gradient of a given node's representation in a factorisation-based model is a sum of gradients obtained from the respective facts in which said node appears, and its overall gradient also includes a global component resulting from facts it is not connected to. Hence, once can consider the gradient from facts in the knowledge graph as messages, their sum as the aggregation operation, and the combine/update operation as the gradient descent update step (which includes the additional global component). Based on this model design, the work then applies the model to inductive and transductive baselines, achieving very strong performance, particularly in the latter inductive setting.

### Strengths ###
- The intuition of using GD as a bridge between factorisation models and GNN is clever and elegant.
- The empirical results of the ReFACTOR GNN model are strong, and the choices of experimental setup are justified: I particularly enjoyed the call for full ranking evaluation.
- The presentation of the paper is clear and the flow is easy to follow overall.

### Weaknesses ###
I see no major weaknesses in the current work. However, I do have some minor suggestions to improve the work:

- I find that the theoretical understanding of the model's expressive power could be better studied. In particular, this model uses a simple aggregation and update, but includes a global component. This already goes beyond a standard GNN, and thus the 1-WL correspondence does not necessarily apply. Moreover, the ReFACTOR-GNN global component n[v], which considers nodes not connected to v, and thus feels (at least, at a high level) like a global readout. Hence, would existing results and logical characterisations on GNNs with global readouts [1] be potentially applicable to this model? Or does this model go beyond a readout? Adding such discussions would be quite interesting in my opinion.

- Some aspects of the training setup were harder to follow on Page 4. I would recommend discussing the reasons behind this setup before presenting the technical aspects, to make the connection and choices clearer and more natural.

### Recommendation ###

The contributions of this extended abstract are very interesting and its empirical results are strong. I therefore think this work is a clear fit for the Extended Abstract track of LoG. I recommend a Clear Accept.

### Further Feedback ###
Some minor typos:
- Page 4: Subparagraph -> subgraph
- Page 8: There are ?? references. Please add the missing references.

---

### Official Review · Reviewer_KfAe · 2022-10-22

**Overall Score:** 8
**Confidence:** 3

**Review:**

The paper proposes a new family of architectures, REFACTOR GNNS, that interpolates between FMs and GNNs and allows FMs to be used inductively. Specifically, by using a message-passing formalism, the authors show how FMs can be cast as GNNs by reformulating the gradient descent procedure as message-passing operations. Numerically, the proposed method achieves state-of-the-art inductive performance across the board and comparable transductive performance to FMs with much fewer parameters.

Originality: The idea of reformulating the gradient descent procedure as message-passing operations to interpolate between FMs and GNNs is novel and interesting.

Quality: The analysis is technically sound. The experimental validation is solid. It would be better if the authors can provide more theory/intuition on why the proposed method performs better than the baseline methods.

Clarity: This paper may need to be reorganized for a 4-page paper. Key experiments results should be present on the main pages. Besides, there are some '(u, v, ?)' and '??' in the paper. Please proofread and correct them.

Significance: The problem is interesting and important to the community.

---

### Official Review · Reviewer_YmgL · 2022-10-23

**Overall Score:** 6
**Confidence:** 4

**Review:**

The paper proposes a new variant of GNNs, REFACTOR GNNs, which combines both FMs and classic GNNs. In particular, REFACTOR GNNs achieve comparable transductive performance of FMs, and state-of-the-art inductive performance while using an order of magnitude fewer parameters on link prediction tasks.

Advantages:
1. The authors use bridge the gap between FMs and GNNs by their model.
2. The authors address the question of why FMs are stronger multi-relational link predictors compared to plain GNNs.
3. The experimental results are good.
4. In the supplementary section, the authors give some details and proofs, which is commendable.

Disadvantages:
1. The concern about the paper is the lack of results to show. The authors mentioned they follow the standard KGC evaluation protocol by computing two metrics: Mean Reciprocal Ranking (MRR) and Hit Ratios at Top K (Hits@K). However, the figures and tables only show the results of Hit@10. Other results are missing.
2. ‘The Performance vs Parameter Efficiency as #Layers Increases’ part is for comparison with Graph Attention Networks only. Gated Graph Neural Networks and Graph Convolutional Networks can be added as mentioned in Related Work.
3. The code for this paper is not open source.


It seems this paper has been accepted by another conference?

---

### Official Review · Reviewer_ddEq · 2022-10-24

**Overall Score:** 8
**Confidence:** 4

**Review:**

### Paper Summary
The paper argues that factorization machines could be re-interpreted as message passing by looking at the gradient descent training dynamics. Based on this, the paper presents ReFactorGNN, a novel architecture that introduces FM-style lookup to graph-neural networks. Experiments have shown that the ReFactor layer is highly parameter efficient and has state-of-the-art inductive performance on knowledge-graph-completion tasks.

### Strengths
* paper is very well written, clearly motivated, and contains high quality theoretical analysis
* more specifically, section 3 “Implicit Message-Passing in FMs” presents a very fresh & novel take while maintaining theoretical simplicity. I strongly resonate with the claim of “FMs were found to be significantly more accurate than GNNs in KGC tasks, when coupled with specific training strategies,” and using GNNs in KGC is certainly of great importance and interest in the literature.
* strong empirical performance on both transductive and inductive settings.

### Weaknesses
* this paper should be submitted to the full-length track instead of an extended abstract. See below.
* the concept of “infinite message-passing layers” in FMs’ seems important but not discussed in depth enough.

### Recommendation
I recommend acceptance due to the novel contributions this paper has made. There has been very high interest in using GNNs to perform KGC tasks and ReFactorGNN has presented a very elegant solution.

### Questions
* will the code artifacts be open-sourced? I believe researchers in the industry would love to see how this performs in real-world applications.
* how may ReFactor combine & compare with (slightly) newer techniques (e.g. ConvE [2], QuatE [1], etc.)?

### Additional Feedbacks
* potential typo: addend on line 98
* question mark on line 328

### Paper Type
I suggest the author changing the type of submission from the extended abstract to a full submission. There is too much information included in the appendix that is crucial to the presentation of this method, and the main context on its own is not coherent.

[1]: Zhang, Shuai, et al. "Quaternion knowledge graph embeddings." Advances in neural information processing systems 32 (2019).

[2]: Dettmers, Tim, et al. "Convolutional 2d knowledge graph embeddings." Proceedings of the AAAI conference on artificial intelligence. Vol. 32. No. 1. 2018.

---

### Official Review · Reviewer_fWAJ · 2022-10-27

**Overall Score:** 8
**Confidence:** 4

**Review:**

Summary: The paper researches the connection between conventional KG embedding learning methods—specifically DistMult as they call factorization-based models (FMs)— and GNN-based KG embedding learning. They propose RefactorGNN which interpolates FMs towards GNNs with a message-passing interpretation of FMs. This permits RefactorGNN to work in inductive settings unlike FMs—claimed to reach SOTA inductive performance. RefactorGNN is more parameter-efficient compared to FMs or GNNs since the learnable parameters are only the relational embeddings and parameter space does not enlarge with the number of layers. It is memory friendly due to clearing node state cache implementation. Also, it is able to incorporate any existing initial node features in the learning.

Possible improvements: Additional discussion would be appreciated on the relation of RefactorGNN messages to the message functions used in NBFNet study—also inspired by factorisation and translation-based scoring of triples. Also, there is a message-passing interpretation of KG embedding learning in a distributed fashion where message passing of the gradients takes place in “Learning Graph Representations with Embedding Propagation” Garcia-Duran & Niepert, that can be added to the references.

Overall, I vote for accept for the contributions on the message-passing interpretation of conventional KG embedding learning methods since it may open the research path up for further hybrid methods enjoying the advantages of both approaches.

---

### Meta-Review · Area_Chair_rfgK · 2022-11-20

**Confidence:** 5
**Recommendation:** Accept

**Meta Review:**

This is a strong paper that bridges the gap between traditional knowledge-graph embedding methods and GNNs by allowing the former to work in an inductive setting. Moreover, this new model, called ReFactorGNNs, are more parameter efficient than traditional KG embedding approaches. There is a strong agreement among the reviewers regarding the quality and scientific merit of the work.

---

### Decision · Program_Chairs · 2022-11-22

Accept (Poster)